bioinformatics/evolution

karyotype, chromosomes, X0 sex-determination system, SNPs, coverage, water fleas

**Authors for correspondence:**
Luca Cornetti
e-mail: luca.cornetti@unibas.ch
Dieter Ebert
e-mail: dieter.ebert@unibas.ch

# No evidence for genetic sex determination in *Daphnia magna*

## Luca Cornetti and Dieter Ebert

Department of Environmental Sciences, Zoology, University of Basel, 4051, Basel, Switzerland

 LC, 0000-0002-7188-4048; DE, 0000-0003-2653-3772

Mechanisms of sex determination (SD) differ widely across the tree of life. In genotypic sex determination (GSD), genetic elements determine whether individuals are male or female, while in environmental sex determination (ESD), external cues control the sex of the offspring. In cyclical parthenogens, females produce mostly asexual daughters, but environmental stimuli such as crowding, temperature or photoperiod may cause them to produce sons. In aphids, sons are induced by ESD, even though GSD is present, with females carrying two X chromosomes and males only one (X0 SD system). By contrast, although ESD exists in *Daphnia*, the two sexes were suggested to be genetically identical, based on a 1972 study on *Daphnia magna* (2n=20) that used three allozyme markers. This study cannot, however, rule out an X0 system, as all three markers may be located on autosomes. Motivated by the life cycle similarities of *Daphnia* and aphids, and the absence of karyotype information for *Daphnia* males, we tested for GSD (homomorphic sex chromosomes and X0) systems in *D. magna* using a whole-genome approach by comparing males and females of three genotypes. Our results confirm the absence of haploid chromosomes or haploid genomic regions in *D. magna* males as well as the absence of sex-linked genomic regions and sex-specific single-nucleotide polymorphisms. Within the limitations of the three studied populations here and the methods used, we suggest that our results make the possibility of genetic differences among sexes in the widely used *Daphnia* model system very unlikely.

## 1. Introduction

The X0 sex-determination (SD) system (also referred to as XX/X0), in which females have two X chromosomes (XX) and males have only one (X0), is derived from the more common XY chromosomal SD systems [1]. The X0 SD system likely evolved from the XY system when the Y chromosome was lost [2–4],

which happened independently in several taxa. The X0 SD system is observed in nematodes [5], insects [6], crustaceans [7], gastropods [8], fish [9] and even in a few mammals [10]. Among insects with an X0 SD system are some particular aphid species (i.e. from the genera *Myzus* and *Acyrthosiphon*) that reproduce through cyclic parthenogenesis, in which several generations of asexual reproduction are followed by a generation of sexual reproduction [6]. This happens when environmental cues associated with the onset of winter (i.e. low temperatures and short photoperiods) regulate molecular signalling cascades that stimulate the production of male offspring, which inherit only one copy of the female's X chromosomes [11]. Thus, sex in aphids is determined by both genetic and environmental factors.

Crustaceans of the genus *Daphnia* also usually reproduce by cyclic parthenogenesis with environmental cues stimulating the production of sons [12]. Under favourable conditions, *Daphnia* reproduce asexually, i.e. mothers produce female offspring through diploid (2n) parthenogenetic eggs, which are placed in the brood chamber and hatch after 3 days. Towards the end of the growth season, prompted by a complex set of stimuli including population density, day length and temperature [13], mothers may switch to producing haploid resting eggs, which are encapsulated in a structure called ephippium. Ephippial eggs require fertilization by males, whose production is, in turn, environmentally stimulated. The resting eggs are able to resist adverse conditions (e.g. low winter temperatures or summer droughts), and daughters hatch from them after favourable environmental cues are perceived [12]. Males and females are both produced asexually and are considered genetically identical, with sexual differentiation being induced by differential gene expression [14,15].

Crustaceans show very diversified SD mechanisms, including environmental sex determination (ESD), genotypic sex determination (GSD), polygenic (i.e. multiple genes with either male- or female-determining effect), and cytoplasmic (i.e. cytoplasmic organelles or parasite control sex development) systems [4,16]. Among crustaceans with GSD, several species have an X0 SD system, including at least two species (*Branchipus vernalis* and *Chirocephalus nankinensis*) in the Branchiopoda class [16] of lower crustaceans that also contains *Daphnia*. *Daphnia* have ESD, with no evidence of sex chromosomes [17], apart from the identification of an incipient W chromosome observed in some populations with females unable to produce sons, so-called nonmale producers [18]. However, an X0 system, as observed in some other cyclic parthenogens, has never been properly tested and, therefore, cannot be excluded. Because chromosomes in *Daphnia* are small and contracted, karyotype differences among sexes are difficult to determine [19]. To our knowledge, the only study that tested for genetic differences between males and females of the same *Daphnia* clone found no such difference [17]; however, this study used only three genetic markers (allozymes), which allowed it to exclude a haplo-diploid system (males being haploid), but not an X0 SD system, as these markers may have been located on autosomes. *D. magna* has 10 chromosomes (1n). The results from [20], where a male-specific genetic map was constructed, suggested the absence of haploid chromosomes in males of another *Daphnia* species (*D. pulex*, 2n=24), but was not able to exclude the presence of short sex-linked genomic regions.

With the availability of high-quality *Daphnia* genome assemblies [21,22], it is now possible to tailor genomic analyses that can uncover potential differences in the ploidy level of genomic regions or chromosomes between the sexes. Because sex chromosomes differ from autosomes in several ways—zygosity (e.g. hemizygosity), sex-specific patterns of inheritance and reduced or suppressed recombination [23]—they are thus subject to different evolutionary forces and must be known and distinguished from autosomes when asking questions about the genetics and evolution of a study organism.

The similarities between the life cycles of some aphid species and *Daphnia* and the fact that the X0 SD system is widespread among animals, including crustaceans, prompted our interest to test also in *Daphnia*, whether an X0 system contributes to SD. Here, we analysed three *D. magna* genotypes using high-throughput whole-genome sequencing to assess, if this species has an X0 SD system. In addition, we searched for alternative GSD systems by evaluating the occurrence of potential sex-specific single-nucleotide polymorphisms (SNPs) and sex-linked genomic regions [24]. We believe it is important to confirm or dispute the widely held belief that *Daphnia* does not have a GSD system [15,17]. This question is particularly relevant given the potential consequences that the misinterpreting of genetic and genomic data could produce in this important model system.

## 2. Material and methods

Using the standard laboratory conditions described in [25], we maintained three isofemale lines of naturally collected *D. magna* (Genotype CA-CH-1 from Canada, GPS coordinates: lat: 58.771, long:

−93.851; Genotype NO-M3-1 from Norway, lat: 59.0984, long: 11.198; and Genotype RU-KOR1-1 from Russia, lat: 66.4519, long: 33.799). These genotypes were kept in cultures with approximately 20–40 animals per 400 mL jar, conditions in which these *D. magna* clones readily produce daughters and sons. Males and females, which can be easily distinguished morphologically [12], were separated from each other for further analysis.

We sequenced the entire genome of males and females separately for the three *D. magna* genotypes, aiming at a genome-wide coverage of at least 25 X per sample. Since sex in *Daphnia* offspring is determined during oogenesis, before the eggs are extruded into the brood chamber [26], we excluded all egg-carrying females to avoid possible contamination with male genomic DNA during sequencing. Before DNA extraction, we reduced bacterial DNA by maintaining the living animals for three days (with daily transfer) in an antibiotic solution consisting of Ampicillin, Streptomycin and Tetracycline (Sigma) at a concentration of 50 mg L$^{-1}$ each. We also reduced gut content by feeding the animals 5 mg of fine beads of the gel filtration resin Sephadex G-25 (Sigma-Aldrich) twice a day. Genomic DNA was extracted from about 30 animals per sample using the QIAGEN Gentra Puregene Tissue Kit, including the RNaseA (100 mg ml$^{-1}$; Sigma) digestion step. Whole-genome Illumina paired-end sequencing (read length 125 bp) was performed by the Genomics Facility service platform at the Department of Biosystem Science and Engineering (D-BSSE, ETH) in Basel, Switzerland, on an IlluminaHiSeq 2500.

We used Trimmomatic 0.36 [27] with the following parameters: HEADCROP:5, AVGQUAL:25, LEADING:20, TRAILING:20, SLIDINGWINDOW:5:20, MINLEN:36, to remove adapters and low-quality regions from the raw reads. Trimmed reads (see details in electronic supplementary material, table S1) were aligned to a newly assembled *D. magna* high-quality reference genome of a three times inbred Finnish clone (accession LRGB00000000.1) using BWA 0.7.7 [28]. Our reference genome, which was obtained using high coverage (greater than 100X) PacBio long-read sequencing, had a maximum contig size of almost 9 Mb (essentially a chromosome arm) and an N50 of about 2.5 Mb, consisted of chromosome-level scaffolds generated using JoinMap [29,30] and the RAD-sequencing-based genetic map data of [31,32] (BioProject ID: PRJNA624896; Fields *et al.*, in prep).

To test for a hypothetical X0 SD system, given the expected difference in sex chromosome ploidy between males and females [24], we focused on read coverage to identify sex chromosomes. The BAM files that resulted from the mapping to the reference genome were filtered for mapping quality (-q 30, electronic supplementary material, table S1) with SAMTOOLS 0.1.19 [33] and used to obtain the coverage of each position in the genome for the six analysed samples using BEDTools v. 2.18.1 [34]. After excluding the positions with extremely high coverage (greater than 200 X), we calculated coverage values for non-overlapping sliding windows of different sizes (5 kbp and 10 kbp). Windows with an average coverage lower than 5 X in the female samples were excluded from further analyses because these low coverage values might bias the ratio calculation (see below). We did not apply the same filter to males, since low coverage windows may underlie the occurrence of only one copy of the X chromosome. We standardized the window coverage values between female and male samples by dividing the window coverage with the sex-specific genome-wide average coverage. Then, for each window, we divided the female coverage by the male coverage with the expectation that autosomes should result in a ratio around one and sex chromosomes in a ratio around two (i.e. females have two copies of the X chromosome (XX) and males have only one copy (X0)). Coverage values were plotted for each chromosome using R v. 3.6.2 [35].

In addition to coverage calculation, we performed a variant calling analysis to search for potential sex-linked genomic regions, which may unveil alternative GSD systems. We used the previously obtained BAM files and the HaplotypeCaller function in GATK 4.0 [36] to generate a g.vcf file for each sample. Then, we combined all the g.vcf files and considered high-quality biallelic SNPs satisfying the following conditions: a minimum depth of 10, a maximum depth of 100, a minimum genotype quality of 30 and no missing data allowed. For each SNP, we calculated the $F_{st}$ between females and males with the aim of identifying putative sex-specific polymorphisms. SNP filtering and $F_{st}$ calculation were performed with VCFTOOLS 0.1.16 [37]. We used the Integrative Genome Viewer software (IGV 2.3.92 [38]), and the reference genome and the BAM files as input, to confirm the calling reliability of the SNPs showing association with sex.

# 3. Results

We found an average genome-wide coverage ratio across all chromosomes of 1.002 (standard deviation 0.088) and 1.001 (standard deviation 0.063) for sliding windows of 5 kbp and 10 kbp, respectively. No

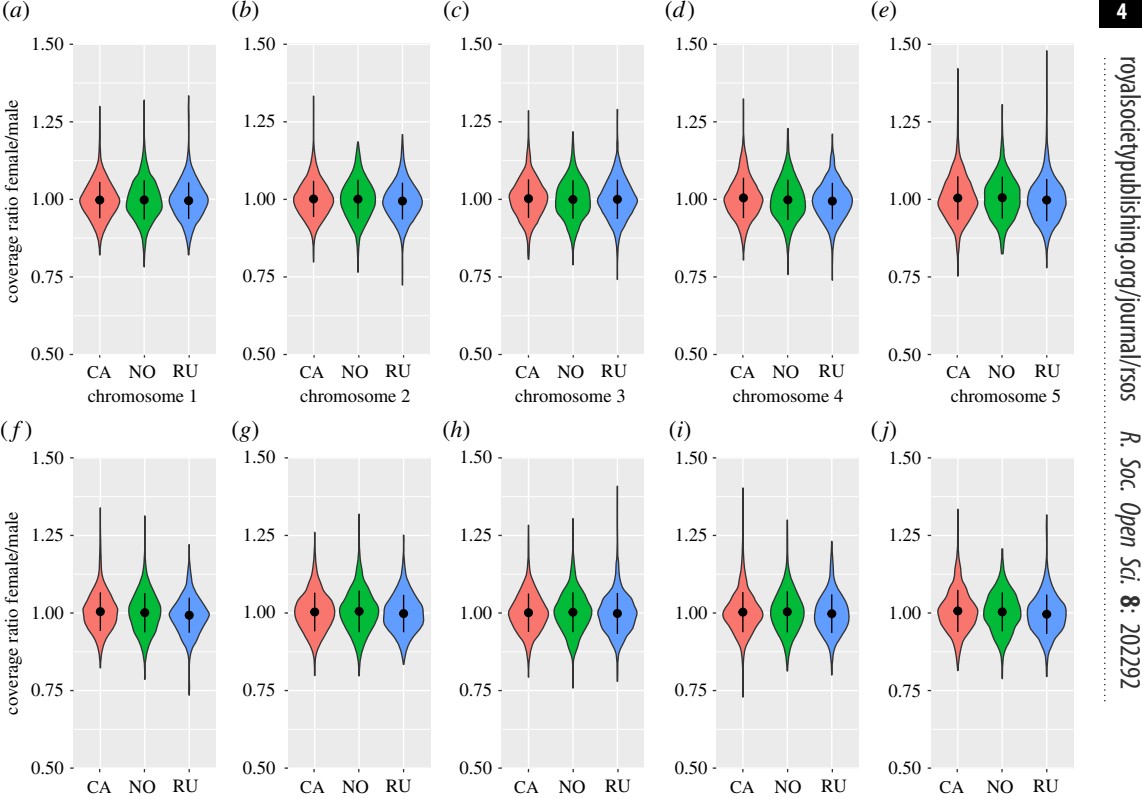

**Figure 1.** Violin plots displaying the whole range of coverage ratio female/male values for all the 10 kbp non-overlapping sliding windows considered. Black dots inside the violin plots represent mean, while black whiskers indicate standard deviation. The 10 chromosomes are shown separately, and the three analysed *D. magna* genotypes are plotted side-by-side (CA: CA-CH-1 from Canada; NO: NO-M3-1 from Norway; RU: RU-KOR1-1 from Russia). Putative sex chromosomes in an X0 sex-determination system should have a coverage ratio close to two, while autosomes have a coverage ratio close to one.

chromosomes nor chromosome sections showed windows with coverage ratio values close to two: all genotypes showed the same pattern, with the coverage ratio in all chromosomes very close to one (figure 1; electronic supplementary material, figures S1 to S6). Given that an incipient W chromosome located in the peri-centromeric region of chromosome 3 was identified in nonmale-producer females, collected from some *D. magna* populations, we particularly focused on this region. Electronic supplementary material, figures S1 to S6 indicate that, also in this location, all the windows had a coverage ratio very close to one in all the genotypes.

A total of 1,173,634 SNPs was retained after quality and depth filtering. Among those, six SNPs showed the highest $F_{st}$ ranking value of 0.5. We did not identify other polymorphisms with an $F_{st}$ higher than 0.25. The six SNPs with differentiation between males and females showed all heterozygosity in one sex and homozygosity (or hemizygosity) in the other. These six SNPs are located on five different chromosomes; two polymorphisms are physically linked and lie on the same contig 10 bp away from each other (electronic supplementary material, tables S2 and S3). We investigated these six polymorphisms to determine their authenticity. The read count and the genotype calling, supporting the two alleles in the six SNPs is reported in electronic supplementary material, table S2. Overall, in the heterozygous SNPs, the number of reads displaying the alternative allele was always accounting for a much smaller proportion of the total read depth (on average 14.8%) than the expected 50% for typical heterozygous genotypes. Most homozygote genotypes included some reads with the minor allele observed in the heterozygotes (electronic supplementary material, table S2). Together, the actual difference between the heterozygotes and the homozygotes with regard to read counts per allele was rather marginal. Not surprisingly, the vcf file, resulting from the GATK variant calling pipeline, pointed to low support for the heterozygous variants (electronic supplementary material, table S3). In conclusion, the six SNPs showing apparent association with sex seem not to be reasonable candidates for a GSD system in *Daphnia*. Rather, it seems that an explanation for the apparent sex difference is that, for some of the regions containing putative

sex-specific SNPs, read mapping was inaccurate as the coverage in these locations appeared to be considerably higher than the average (electronic supplementary material, figures S7 to S11). This was combined with a genotype calling that resulted in different SNPs for the two sexes.

# 4. Discussion

SD mechanisms, including ESD and GSD are highly diverse across biological organisms [4]. To date, *Daphnia magna* is generally believed to have an ESD, with no evidence of sex chromosomes or other forms of GSD. However, ESD does sometimes mask an X0 SD system as in some aphid species, which, like *Daphnia*, are cyclic parthenogens [39]. Here, we tested whether a GSD system in *D. magna* can be found, using a chromosome by chromosome coverage analysis and measures of more regional sex differences. To identify potential sex chromosomes, we analysed genome coverage from short-read Illumina data that exploits the difference in sex chromosome ploidy between females (XX) and males (X0) [24], a method that has been successfully applied in multiple organisms with XY, ZW and X0 SD systems (e.g. [26,33,34]). None of the 10 chromosomes revealed a difference between the sexes. We observed a few instances of randomly distributed genomic windows showing a coverage ratio slightly deviating from one, the expected value for autosomes. However, outlier 10 kb windows rarely exceeded the value of 1.2 (figure 1), and we observed no overlap of such cases among clones. The same conclusion was reached for an analysis of SNPs with sex-specific differences in zygosity. Overall, our results provide no evidence for GSD in *D. magna*.

Many organisms with GDS have homomorphic sex chromosomes that, in contrast with heteromorphic sex chromosomes (e.g. X and Y), are nearly identical in terms of gene content and size [4]. This type of sex chromosome, in which the non-recombining region has not spread significantly beyond the locus responsible for SD, normally shows limited sequence divergence and is challenging to identify [24,40]. Some plants and several animal species including fish, reptiles, birds and amphibians display no differences between the chromosomal sets of male and female genomes, but sex is genetically determined by single or multiple sex-linked genomic regions that contain SNPs significantly associated with the phenotypic sex (e.g. [40–44]). Our experimental setting allowed us to assess the presence of sex-specific SNPs by comparing the genotypes of the two sexes. Using an $F_{st}$ approach, we identified six SNPs potentially associated with the phenotypic sex, where all three individuals of one sex were heterozygous and all three individuals of the other sex were homozygous. We manually reviewed the six-candidate SNPs to determine their authenticity and observed low reliability for all candidate polymorphisms. Overall, if one were to consider 'true' heterozygous SNPs only polymorphisms where the ratio between the numbers of reads supporting the two alleles was between 1 and 1.5, none of the candidate variants would still be included in the SNP list. Furthermore, for four of the six SNPs, the females were homozygote and the males were heterozygote (electronic supplementary material, table S2), which is difficult to bring in line with GSD in a cyclic parthenogen, where males are only rarely produced (a heterozygous male is very unlikely to originate by asexual reproduction from a homozygous mother). This analysis confirms the low reliability of the identification of heterozygous polymorphisms using next-generation sequencing data for cases where specific sites are targeted [45].

Our analyses did not reveal any SNP being significantly associated with the phenotypic sex in our samples. It should be noted, however, that our analysis assumes that, if a SNP would be responsible for GSD in *D. magna*, it is the same polymorphism in the three genotypes, originating from the three populations studied here. It has been reported, however, that in some species, different populations show different mechanisms of SD (e.g. [46,47]. Therefore, if in *D. magna* SNPs explaining GDS vary among populations, their detection would require a different sampling scheme. Another limitation of our study, concerning the identification of sex-specific SNPs, could be related to the absence of the sex-determining region(s) in the reference genome. For example, as observed for most large eukaryotic genomes [48], the assembly quality is typically lower around the centromeres which may result in missing sections containing the actual sex-determining region. Similarly, in case of high-sequence divergence and/or major structural variation between the reference genome and the region responsible for SD, SNPs associated with the phenotypic sex might be overlooked because of unreliable read mapping to the reference genome [49].

Xu *et al*.'s (2015) study on *D. pulex* (2n=24) [20]—where a male-specific genetic map was constructed by sequencing the whole genome of individual sperms—pointed to a similar conclusion as ours, although this study was not designed to test whether some forms of GSD occur in *Daphnia*. Their

genetic map [20] was based on the assumption that sperms are produced by meiosis and, thus, chromosomes are subject to crossover. If an X0 SD system were present in *D. pulex*, the approach would have failed for the putative X chromosome, as no crossover would have occurred and all markers on the putative X chromosome would have been perfectly linked. However, although the male-specific *D. pulex* genetic map presented by [20] suggested that all male chromosomes are diploid, it revealed blocks where several markers were in perfect linkage, which would be consistent with sex-determining regions in the genome. Our study ruled out the hypothesis of haploid sex-linked genomic regions which may result, for example, from autosome-ancestral sex chromosome fusions [50].

In an earlier study, a female-determining incipient W chromosome was described in some *D. magna* populations [18]. Some individuals of these populations cannot produce males, neither under natural nor artificial (i.e. when exposed to hormones) conditions. The locus determining the nonmale-producer trait follows Mendelian inheritance and has been mapped to a 3.6 Mb peri-centromeric region of chromosome 3 that includes genes involved in sex differentiation [18]. The *D. magna* populations analysed in our study do not overlap with the nonmale-producer populations analysed in [18] and do not have, to the best of our knowledge, nonmale-producers. However, given the hypothesized ZW sex chromosome system associated with chromosome 3, we paid close attention to this region. Our results indicate that the peri-centromeric region of chromosome 3 does not show anomalous patterns: we identified neither haploid genomic widows nor sex-linked SNPs in this region, suggesting that the putative ZW system is specific to *D. magna* nonmale-producer populations.

# 5. Conclusion

Our study has revealed that *D. magna* has no sex chromosomes. Together with the evidence in [20], this finding suggests that the absence of an X0 SD system may be more widely spread within the genus *Daphnia*. With our whole-genome sequencing approach, we also did not find evidence for other forms of GSD in *D. magna* by showing an absence of sex-linked genomic regions or SNPs. Our methods cannot fully rule out alternative forms of GSD (e.g. population-specific GSD) or some forms of artefacts (e.g. an incomplete reference genome being used), but we consider this unlikely. Our findings are relevant for our understanding of the *Daphnia* model system, which is intensively used in ecological, genetic and evolutionary studies [51,52].

Data accessibility. Data and codes associated with this study are provided on the Dryad Digital Repository: https://doi. org/10.5061/dryad.tqjq2bvz3.

Authors' contributions. L.C. and D.E. conceived of the study. L.C. collected and analysed the data. L.C. wrote the first draft of the manuscript, and all authors provided critical review and approved its final version. All authors agree to be held accountable for the content.

Competing interests. The authors declare no competing interests.

Funding. This study was supported by the Swiss National Science Foundation (grant no. 310030_188887) and the University of Basel.

Acknowledgement. We thank Jürgen Hottinger, Michelle Krebs and Urs Stiefel for assistance in the laboratory. DE thanks Nancy Moran for an inspiring discussion on SD in cyclic parthenogens. We are grateful to all the members of the Ebert research group who made helpful comments on the manuscript. Suzanne Zweizig improved the language of the text.

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
