## [Peer Review File · Royal Society Open Science]

Review History

RSOS-202292.R0 (Original submission)

Review form: Reviewer 1

Is the manuscript scientifically sound in its present form?

Yes

Are the interpretations and conclusions justified by the results?

Yes

Is the language acceptable?

Yes

Do you have any ethical concerns with this paper?

No

Have you any concerns about statistical analyses in this paper?

No

Recommendation?

Accept as is

Comments to the Author(s)

The authors bravely modified this manuscript.

Review form: Reviewer 2

Is the manuscript scientifically sound in its present form?

No

Are the interpretations and conclusions justified by the results?

No

Is the language acceptable?

Yes

Do you have any ethical concerns with this paper?

No

Have you any concerns about statistical analyses in this paper?

No

Recommendation?

Reject

Comments to the Author(s)

Unfortunately, the present study has an unavoidable weak point associated with such comprehensive screening study. Namely, authors should discuss based only on the found things. Should not discuss based on the things NOT found in comprehensive study because "comprehensive" is not "complete". If authors could find a sex-biased chromosome or genomic regions, the discussion about the possibility of GSD in daphnids was proper. Here authors constructed this paper based on the result in which authors could not find sex-biased regions. In such case, reviewers and readers cannot judge validity of discussion due to enormous possibilities other than author's suggestion (and less information about experimental procedure). For example, focal daphnids have a GSD region but reference genome does not cover it. Another example, GSD region is highly diverged and not mapped on the reference. These are no more than a single example. I repeatedly argue that authors should not discuss based on the things NOT found in comprehensive study.

Review form: Reviewer 3

Is the manuscript scientifically sound in its present form?

Yes

Are the interpretations and conclusions justified by the results?

Yes

Is the language acceptable?

Yes

Do you have any ethical concerns with this paper?

No

Have you any concerns about statistical analyses in this paper?

No

Recommendation?

Accept with minor revision (please list in comments)

Comments to the Author(s)

I read the manuscript of Cornetti and Ebert, entitled “No evidence for genetic sex-determination in *Daphnia*”, with great interest. The authors inspired from the unique XX/X0 sex determination system of aphids, where both sexual and asexual (cyclic parthenogenesis) reproduction can occur, they investigated *Daphnia magna*, another arthropod with cyclic parthenogenesis, for the presence of sex chromosomes.

The methodology is straightforward and to the degree that I can comment for the available information, was correctly applied. The authors used three population of *D. magna* and they sequenced the genomes of males and females independently. Subsequently, they applied a comparative genome coverage (read depth) and a SNPs analysis, to check sex-specific regions/SNPs. The authors conclude that they did not detect any sex-specific difference between male and female genomes, which indicates the absence of sex chromosomes, and therefore GSD.

I completely agree with a previous reviewer that the absence of sex-specific regions revealed by genome coverage or SNPs distribution, is not absolute support of lack of GSD, as in some rare cases (e.g. documented in fish lineages), recombination can stop in a tiny genomic region, where few SNPs are sufficient to create a sex-specific region and give rise to a novel sex-determining locus. This scenario is correctly mentioned in the discussion.

Furthermore, the authors examined only *Daphnia magna*, therefore, the title should be more specific and be rephrased as “No evidence for genetic sex-determination in *Daphnia magna*”.

One controversial matter is the previous report of ZZ/ZW system in *D. magna*, recently reported by Reisser et al. (<https://academic.oup.com/mbe/article/34/3/575/2632624>). Since the authors most probably examined different population, perhaps should be mentioned in a paragraph in the discussion, that although the sex determination of *Daphnia* deserves more exploration as we are still lacking the exact mechanism that controls the sex differentiation, it is not uncommon in some species, different populations to show different modes of sex determination. For example, in the frogs *Xenopus tropicalis* (<https://pubmed.ncbi.nlm.nih.gov/26216983/>) and *Rana rugosa* (<https://pubmed.ncbi.nlm.nih.gov/22143254/>).

Beyond these comments, I believe that this study brings new information on sex determination in a model species, and therefore deserves to be published in open Science, after minor revisions.

Decision letter (RSOS-202292.R0)

Dear Dr Cornetti

The Editors assigned to your paper RSOS-202292 "No evidence for genetic sex-determination in *Daphnia*" have now received comments from reviewers and would like you to revise the paper in accordance with the reviewer comments and any comments from the Editors. Please note this decision does not guarantee eventual acceptance.

Please submit your revised manuscript and required files (see below) no later than 21 days from today's (ie 03-Feb-2021) date. Note: the ScholarOne system will 'lock' if submission of the revision is attempted 21 or more days after the deadline. If you do not think you will be able to meet this deadline please contact the editorial office immediately.

on behalf of Prof Kevin Padian (Subject Editor)
openscience@royalsociety.org

Associate Editor Comments to Author:

Thank you for transferring your manuscript to Royal Society Open Science from Biology Letters. Your efforts to revise your paper are appreciated, though Reviewer 2 (in particular) and Reviewer 3 offer commentary that you need to provide responses to and consider whether your paper needs further amendments as a consequence. The editors would (ideally) like you to address the concern raised by reviewer 2 in relation to inference and comprehensive vs completeness to a

greater extent - as RSOS does not have a page limit, you have the luxury of expanding the manuscript to ensure these matters are appropriately addressed.

Editor comments:

Thank you for your submission. Following the comments of the AE, I ask you to attend closely to the remaining issues expressed by the reviewers. It is critical to address these fully or we cannot publish the manuscript. You may of course disagree but the sides have to be laid out fairly and carefully. We look forward to your revised submission and if you need more time please let us know. Best wishes.

Reviewer comments to Author:

Reviewer: 1

Comments to the Author(s)

The authors bravely modified this manuscript.

Reviewer: 2

Comments to the Author(s)

Unfortunately, the present study has an unavoidable weak point associated with such comprehensive screening study. Namely, authors should discuss based only on the found things. Should not discuss based on the things NOT found in comprehensive study because "comprehensive" is not "complete". If authors could find a sex-biased chromosome or genomic regions, the discussion about the possibility of GSD in daphnids was proper. Here authors constructed this paper based on the result in which authors could not find sex-biased regions. In such case, reviewers and readers cannot judge validity of discussion due to enormous possibilities other than author's suggestion (and less information about experimental procedure). For example, focal daphnids have a GSD region but reference genome does not cover it. Another example, GSD region is highly diverged and not mapped on the reference. These are no more than a single example. I repeatedly argue that authors should not discuss based on the things NOT found in comprehensive study.

Reviewer: 3

Comments to the Author(s)

I read the manuscript of Cornetti and Ebert, entitled "No evidence for genetic sex-determination in *Daphnia*", with great interest. The authors inspired from the unique XX/X0 sex determination system of aphids, where both sexual and asexual (cyclic parthenogenesis) reproduction can occur, they investigated *Daphnia magna*, another arthropod with cyclic parthenogenesis, for the presence of sex chromosomes.

The methodology is straightforward and to the degree that I can comment for the available information, was correctly applied. The authors used three population of *D. magna* and they sequenced the genomes of males and females independently. Subsequently, they applied a comparative genome coverage (read depth) and a SNPs analysis, to check sex-specific regions/SNPs. The authors conclude that they did not detect any sex-specific difference between male and female genomes, which indicates the absence of sex chromosomes, and therefore GSD.

I completely agree with a previous reviewer that the absence of sex-specific regions revealed by genome coverage or SNPs distribution, is not absolute support of lack of GSD, as in some rare cases (e.g. documented in fish lineages), recombination can stop in a tiny genomic region, were

few SNPs are sufficient to create a sex-specific region and give rise to a novel sex-determining locus. This scenario is correctly mentioned in the discussion.

Furthermore, the authors examined only *Daphnia magna*, therefore, the title should be more specific and be rephrased as “No evidence for genetic sex-determination in *Daphnia magna*”.

One controversial matter is the previous report of ZZ/ZW system in *D. magna*, recently reported by Reisser et al. (<https://academic.oup.com/mbe/article/34/3/575/2632624>). Since the authors most probably examined different population, perhaps should be mentioned in a paragraph in the discussion, that although the sex determination of *Daphnia* deserves more exploration as we are still lacking the exact mechanism that controls the sex differentiation, it is not uncommon in some species, different populations to show different modes of sex determination. For example, in the frogs *Xenopus tropicalis* (<https://pubmed.ncbi.nlm.nih.gov/26216983/>) and *Rana rugosa* (<https://pubmed.ncbi.nlm.nih.gov/22143254/>).

Beyond these comments, I believe that this study brings new information on sex determination in a model species, and therefore deserves to be published in open Science, after minor revisions.

===PREPARING YOUR MANUSCRIPT===

===PREPARING YOUR REVISION IN SCHOLARONE===

To revise your manuscript, log into <https://mc.manuscriptcentral.com/rsos> and enter your Author Centre - this may be accessed by clicking on "Author" in the dark toolbar at the top of the

page (just below the journal name). You will find your manuscript listed under "Manuscripts with Decisions". Under "Actions", click on "Create a Revision".

Author's Response to Decision Letter for (RSOS-202292.R0)

See Appendix A.

RSOS-202292.R1 (Revision)

Review form: Reviewer 2

Is the manuscript scientifically sound in its present form?

No

Are the interpretations and conclusions justified by the results?

Yes

Is the language acceptable?

Yes

Do you have any ethical concerns with this paper?

No

Have you any concerns about statistical analyses in this paper?

No

Recommendation?

Accept as is

Comments to the Author(s)

The manuscript has been revised appropriately.

Decision letter (RSOS-202292.R1)

Dear Dr Cornetti,

It is a pleasure to accept your manuscript entitled "No evidence for genetic sex-determination in *Daphnia magna*" in its current form for publication in Royal Society Open Science.

on behalf of Prof Kevin Padian (Subject Editor)
openscience@royalsociety.org

Editor comments:

Thanks for your attention to the revisions. We are pleased to accept your manuscript as is. Best wishes.

Reviewer comments to Author:

Reviewer: 2

Comments to the Author(s)

The manuscript has been revised appropriately.

Appendix A

Dear Editor,

We thank you and the reviewers for the constructive comments and the opportunity to further improve our manuscript. Please find below a detailed description of how we have incorporated the reviewers' comments in the revised manuscript. These changes helped to further clarify and strengthen the manuscript and we hope that you will find it suitable for publication in Royal Society Open Science in its revised form.

Our replies are indicated by: ">>"

Sincerely yours,

Luca Cornetti and Dieter Ebert

Associate Editor Comments to Author:

Thank you for transferring your manuscript to Royal Society Open Science from Biology Letters. Your efforts to revise your paper are appreciated, though Reviewer 2 (in particular) and Reviewer 3 offer commentary that you need to provide responses to and consider whether your paper needs further amendments as a consequence. The editors would (ideally) like you to address the concern raised by reviewer 2 in relation to inference and comprehensive vs completeness to a greater extent - as RSOS does not have a page limit, you have the luxury of expanding the manuscript to ensure these matters are appropriately addressed.

>> We addressed all the concerns raised by the two Reviewers. In particular, following the suggestions of Reviewer 2, we toned down our conclusions by acknowledging some limitations in our study. We modified the abstract (L25-27), the discussion (by including a new paragraph where we describe alternative explanations to the absence of Genetic Sex Determination, L239-252) and the conclusion (L287-289).

Editor comments:

Thank you for your submission. Following the comments of the AE, I ask you to attend closely to the remaining issues expressed by the reviewers. It is critical to address these fully or we cannot publish the manuscript. You may of course disagree but the sides have to be laid out fairly and carefully. We look forward to your revised submission and if you need more time please let us know. Best wishes.

Reviewer comments to Author:

Reviewer: 1

Comments to the Author(s)

The authors bravely modified this manuscript.

>> Thank you!

Reviewer: 2

Comments to the Author(s)

Unfortunately, the present study has an unavoidable weak point associated with such comprehensive screening study. Namely, authors should discuss based only on the found things. Should not discuss based on the things NOT found in comprehensive study because “comprehensive” is not “complete”. If authors could find a sex-biased chromosome or genomic regions, the discussion about the possibility of GSD in daphnids was proper. Here authors constructed this paper based on the result in which authors could not find sex-biased regions. In such case, reviewers and readers cannot judge validity of discussion due to enormous possibilities other than author’s suggestion (and less information about experimental procedure). For example, focal daphnids have a GSD region but reference genome does not cover it. Another example, GSD region is highly diverged and not mapped on the reference. These are no more than a single example. I repeatedly argue that authors should not discuss based on the things NOT found in comprehensive study.

>> We acknowledge that we did not mention further alternative explanations to the absence of GSD identified with our approaches. Following Reviewer 2’s suggestions we modified the discussion (L239-252), the abstract (L25-27) and the conclusion (L287-289) and mentioned examples of possibilities that we might have missed with our methodology. We wrote a new paragraph where we included the limitations of our study. For example, we made clearer (as also requested by Reviewer 3) that different populations might have different mechanisms of SD, as it is reported for some other species. We also specified that the reference genome might lack the actual sequence responsible for sex determination and mention that high sequence divergence and/or structural variation between the reference genome and the sex-determining region might prevent the identification of sex-specific SNPs (L239-252).

Reviewer: 3

Comments to the Author(s)

I read the manuscript of Cornetti and Ebert, entitled “No evidence for genetic sex-determination in *Daphnia*”, with great interest. The authors inspired from the unique XX/X0 sex determination system of aphids, where both sexual and asexual (cyclic parthenogenesis) reproduction can occur, they investigated *Daphnia magna*, another arthropod with cyclic parthenogenesis, for the presence of sex chromosomes.

The methodology is straightforward and to the degree that I can comment for the available information, was correctly applied. The authors used three population of *D. magna* and they sequenced the genomes of males and females independently. Subsequently, they applied a comparative genome coverage (read depth) and a SNPs analysis, to check sex-specific regions/SNPs. The authors conclude that they did not detect any sex-specific difference between male and female genomes, which indicates the absence of sex chromosomes, and therefore GSD.

I completely agree with a previous reviewer that the absence of sex-specific regions revealed by genome coverage or SNPs distribution, is not absolute support of lack of GSD, as in some rare cases (e.g. documented in fish lineages), recombination can stop in a tiny genomic region, where few SNPs are sufficient to create a sex-specific region and give rise to a novel sex-determining locus. This scenario is correctly mentioned in the discussion.

>> *We are glad to hear that our manuscript, including its first revision, has been positively evaluated*

Furthermore, the authors examined only *Daphnia magna*, therefore, the title should be more specific and be rephrased as “No evidence for genetic sex-determination in *Daphnia magna*”.

>> *We agree and we modified the title accordingly*

One controversial matter is the previous report of ZZ/ZW system in *D. magna*, recently reported by Reisser et al. (<https://academic.oup.com/mbe/article/34/3/575/2632624>). Since the authors most probably examined different population, perhaps should be mentioned in a paragraph in the discussion, that although the sex determination of *Daphnia* deserves more exploration as we are still lacking the exact mechanism that controls the sex differentiation, it is not uncommon in some species, different populations to show different modes of sex determination. For example, in the frogs *Xenopus tropicalis* (<https://pubmed.ncbi.nlm.nih.gov/26216983/>) and *Rana rugosa* (<https://pubmed.ncbi.nlm.nih.gov/22143254/>).

>> *Thank you for the suggestion. We specified more precisely that the populations analyzed in our study do not overlap with the (rare) populations studied by Reisser et al., (2017) (L273-274). As suggested, we also highlighted that in some species, different populations show different modes of sex determination (L242-244)*

Beyond these comments, I believe that this study brings new information on sex determination in a model species, and therefore deserves to be published in open Science, after minor revisions.

>> *Thank you!*